# Heritability and Trends in Selected Udder Traits and Their Relation to Milk Production in Holstein-Friesian Cows

**DOI:** 10.3390/ani15091276

**Published:** 2025-04-30

**Authors:** Zsolt Jenő Kőrösi, Szabolcs Albin Bene, László Bognár, Ferenc Szabó

**Affiliations:** 1National Association of Hungarian Holstein Friesian Breeders, Lőportár u. 16, H-1134 Budapest, Hungary; 2Institute of Animal Sciences, Georgikon Campus, Hungarian University of Agriculture and Life Sciences, Deák Ferenc u. 16, H-8360 Keszthely, Hungary; 3Department of Animal Sciences, Albert Kázmér Faculty of Agriculture and Food Sciences, Széchenyi István University, Vár t. 2, H-9200 Mosonmagyaróvár, Hungary

**Keywords:** udder attachment, height, depth, texture, central ligament, teat placement

## Abstract

This paper is concerned with the correlation between the conformation of the udder and the milk yield during lactation in Holstein-Friesian cows. It shows the heritability of the traits studied, their relationship to each other, and their phenotypic and genetic trends. It is noteworthy that while milk yield increased significantly over the ten-year period, udder traits remained almost unchanged. This was despite the fact that milk yield and udder conformation traits were included together in the selection index.

## 1. Introduction

The Holstein-Friesian (HF) breed, which originated in the Netherlands and was developed in North America from 1800s, is undoubtedly the most recognized and widespread dairy breed in the world [1]. It has been bred in Hungary since the 1970s and has since become the largest breed in terms of numbers [2]. The main role of the HF breed is to produce high milk yield, so among the many other requirements, proper udder conformation is important for milk production [3]. On the other hand, as the udder of the HF dairy cow, like that of other breeds, has the primary function of milk production and is also important for health and longevity, there are strict requirements for its conformation [4,5]. The demands on the cow’s udder and the criteria for udder evaluation and its inclusion in the selection program have changed as milk yields have increased and milking methods have evolved [3,6,7,8,9].

However, selection for udder conformation can also have negative effects because the udder is associated with cow health, longevity, and milk yield [3,4,5]. Therefore, these relationships and genetic determination should be monitored regularly to avoid adverse effects. Several studies have reported the relationship between udder conformation and health status [4,10,11,12,13,14,15,16]. There are publications reporting an association between udder morphology and longevity in dairy cows. The explanation for this is that if the udder develops unfavourably, the cow’s milk production is lower and the risk of mastitis is higher, so the cows are culled earlier and their productive life is shorter [4,5,17,18,19,20,21].

There is an association between some udder conformation traits and milk yield in cows due to some biological mechanisms [4,22,23,24]. Milk secretion and milk yield depend on udder elasticity, blood supply, and glandular composition, which are related to udder size and conformation [4,22]. The same udder diseases are also related to milk production [4,22,23]. Deeper udders are associated with an increased incidence of mastitis, resulting in lower milk yield. Higher yielding cows tend to have larger udders [4,23,24]. According to White and Vinson [4] there was a positive correlation between some udder measurements and milk yield. Buchanan et al. [22] and Damron [23] reported weak phenotypic and genetic correlation coefficient values between some udder conformation traits and milk yield. Khan and Khan [24] reported high positive genetic association between udder measurements; however, correlation of 305-day milk yield with udder length, udder width, UD, and udder circumference were weak or moderate. Tilki et al. [25] found that 305-day milk yield had a positive correlation with the distance between the anterior and posterior teats and the pre-milking udder scores. Their conclusion was that teat and udder measurements and mammary scores had a significant influence on milk yield. Deng et al. [26] found similar weak or moderate correlation coefficients between milk yield and udder circumference, length, fore UD, rear UD, and udder capacity. Saleh et al. [27] reported that the genetic relationship of udder width with total milk yield and persistency was strong and positive. However, they found strong negative genetic correlations between udder width and peak milk yield and lactation length, and rear teat distance with persistency. According to Mederios et al. [16], the strongest genetic correlations were observed between front teat distance, rear teat distance and UD. As the relationship between milk production and udder conformation is not clear in the literature, and there are conflicting opinions among some authors, it is important to study this relationship breed by breed and herd by herd. It is also important to have a clear understanding of the genetic determination of different udder traits in order to optimize selection criteria.

According to the literature, the heritability (*h*^2^) estimates for the most important udder conformation and milk yield traits are low or medium and similar for each order or slightly lower for udder conformation [28,29,30,31,32,33]. Mederios et al. [16] have published medium and high, and Khan and Khan [24] high *h*^2^ estimates for several udder traits. Therefore, the simultaneous selection response on both milk yield and udder conformation depends on the *h*^2^ of the two groups of traits, and also on the relationships between them [28,29,30]. These literature sources report different *h*^2^ values for udder traits. Therefore, we believe that a re-evaluation is needed in this area as well.

As milk yield increases, udder conformation can change to varying degrees over time. Theron and Mostert [34] found that in South African HF cattle, all udder traits showed an increasing genetic trend, except for teat length, and a slightly positive trend for UD. This result does not agree with the fact that UD has a positive extreme and front teat attachment, and rear udder height (RUH) have a decreasing genetic trend. On the other hand, front teat placement (FTP) and rear teat placement had an increasing trend similar to those obtained by Dube et al. [35]. The genetic trends obtained for these traits are probably due to selection for milk yield [35]. Therefore, they believe it is more reliable to monitor the traits of the progeny in each generation and then evaluate the udder and milk traits over several generations. Ismael et al. [36] have found that most of the udder traits have an increasing genetic trend, except the fore udder attachment (FU).

The aforementioned literary sources also show that it is important to continuously examine the system of udder evaluation criteria. Several scientists have pointed out that morphological evaluation and scoring of cows’ udders is essential [18,19,32,37]. However, there are modern genome-wide association studies for udder traits that suggest that many candidate genes could provide valuable information on the genetic architecture of udder traits [38,39]; the traditional udder scoring practice will retain its importance in the future [31]. Considering that there are discrepancies and contradictions in the literature, and that udder conformation also depends on herds and breeds, continuous genetic evaluation is an important task from a development perspective.

In Hungary, within the current scoring system following the guidelines of World Holstein Friesian Federation (WHFF) [3], there are 21 linear traits, 10 of them for udder, i.e., FU, FTP, teat length, UD, RUH, central ligament (CL), rear teat placement, rear udder width, udder texture (UT), and udder score. In the selection index, besides production and number of functional traits, there are four udder traits, CL, FTP, UD, and teat placement, as an udder composite index [2].

Based on the discrepancy cited above, this study aimed to evaluate the *h*^2^ estimates of some important udder conformation traits (FU, RUH, CL, UD, FTP, and UT), their relationship with each other and with production yield, as well as their phenotypic and genetic trends over a 10-year period in HF cows within a relatively high milk yield level. We expect that the results will help to improve the udder scoring system and reconsider the inclusion of some of the udder traits in the HF selection program.

## 2. Materials and Methods

### 2.1. Cow Population, Their Management, Data, and Traits

The data from a total of 15,032 HF cows were used for this study. The cow population came from 6 herds, offspring of 666 sires and 11,787 dams, born between 2008 and 2018 in Hungary (Table 1). Cows were housed in a loose free stall system with both a communal lying area and an open lying area. Milking was performed in a milking parlor or by automatic milking systems. Cows were fed on a TMR (Total Mixed Ration) basis throughout the year. Their diet was based on maize silage or silage of other crops and concentrates supplemented with protein sources, minerals, and vitamins. The cows’ daily ration depended on their daily MY, condition, and biological status. 

The registered phenotypic performance data of the animals were extracted from the database of the Association of Hungarian Holstein Breeders (NAHHFB). Production traits were milk yield (MY), fat yield (FY), and protein yield (PY) over 305 days of first lactation according to the INTERBIULL guidelines [40]. Only data from cows with a MY result between 5000 and 18,000 kg were used. The udder conformation of all cows was scored for fore udder attachment (FU), rear udder height (RUH), central ligament (CL), udder depth (UD), front teat placement (FTP), and udder texture (UT). Of the 10 udder traits introduced earlier, the reason why we have chosen the previous 6 mentioned udder traits is that according to several literary sources [4,16,22,23,24,25,26], they have the greatest influence on milk production. Scoring was performed by the same association specialist on a linear 1–9 udder score scale [2,3].

### 2.2. Combination of Statistical Methods and Purpose

For our study, in addition to the phenotypic data, we wanted to see the genetic, i.e., breeding value (BV) data for each trait of each animal. The genetic, i.e., BV of the traits studied for each animal was based on random and fixed effects estimated using the BLUP Animal model. As the BLUP requires variance components, these were obtained for the traits using the REML method. When the BV of each trait was available for each animal, the genetic variance could also be estimated. Phenotypic data allowed the investigation of phenotypic correlations and phenotypic trends, while BVs allowed the investigation of genetic variance, correlations, and genetic trends of the traits. From the genetic variance and total genetic variance, it was possible to estimate the *h*^2^ of the traits studied.

### 2.3. Descriptive Statistic

Prior to the model calculations, the descriptive statistical parameters (mean, standard deviation, etc.) of the studied traits were estimated. These were calculated using simple, descriptive mathematical methods using the SPSS 27.0 [41] software (Descriptive statistic module).

### 2.4. Estimation of Variance Components Using the REML Method

The BV estimation with BLUP animal model requires estimation of the variance components. Therefore, these parameters were determined first in our study.

For the variance components and genetic parameters estimation of the analyzed traits, the maximum likelihood (REML) approach was used through the DFREML 3.0 [42] software. Similar to Djedovic et al. [43], the effect of fixed factors on the examined traits was tested using the step-by-step method for the models used in this research. Only statistically significant effects were included in the mentioned procedure. 

Because of constraints and limitations, herd, year of cow birth, season of cow birth, age for conformation traits or age at first calving, and production traits were considered as fixed effects, while the individual (cow) was considered as a random effect. Four groups of cows were created based on cow age at conformation scoring (24.0–27.0, 27.1–30.0, 30.1–33.0, and 33.1–36.0 months). There were also four groups based on age at first calving (20.0–23.0, 23.1–25.0, 25.1–27.0, and 27.1–34.0 months).

There are several non-genetic factors such as nutrition and milking methods or production systems that influence udder characteristics and production. These effects were included in the herd effect, which was considered as a fixed effect in the used model. The structure of the used models is summarized in Table 2.

Based on the above, model 1 (Equation (1)) was used for the calculations in the case of udder conformation traits (FU, RUH, CL, UD, FTP, and UT). For production traits (MY, FY, and PY), model 2 (Equation (2)) was used for the estimation. (1)$$\mathrm{Model}\;1:\;{\hat{y}}_{hijko}=\mu +F_{h}+Y_{i}+M_{j}+S_{k}+a_{o}+e_{hijko}$$(2)$$\mathrm{Model}\;2:\;{\hat{y}}_{hijko}=\mu +F_{h}+Y_{i}+M_{j}+C_{k}+a_{o}+e_{hijko}$$
where *ŷ_hijko_* = the phenotypic expression of observed udder or production trait; *μ* = mean of all observations; *F_h_* = fixed effect of herd; *Y*_i_ = fixed effect of birth year of cow; *M_j_* = fixed effect of birth season of cow (as in Table 1); *S_k_* = fixed effect of the age of cow at scoring; *C_k_* = fixed effect of the age of cow at first calving; *a_o_* = random effect of the animal (cow); and *e_hijko_* = random error.

During this study, the significance of the above-mentioned genetic and environmental factors on the analyzed traits was investigated. 

The genetic variance (*σ*^2^*_d_*), the environmental variance (*σ*^2^*_e_*), and the phenotypic variance (*σ*^2^*_p_*) were estimated using the REML method.

From the genetic variance and total genetic variance, it was possible to estimate the *h*^2^ of the traits studied. The *h*^2^ was calculated using the following formula (Equation (3)):(3)$$h^{2}=\frac{{\sigma^{2}}_{d}}{{\sigma^{2}}_{d}+{\sigma^{2}}_{e}}=\frac{{\sigma^{2}}_{d}}{{\sigma^{2}}_{p}}$$
where *h*^2^ = heritability; *σ*^2^*_d_* = genetic variance; *σ*^2^*_e_* = environmental variance; and *σ*^2^*_p_* = phenotypic variance.

### 2.5. Breeding Value Estimation Using BLUP Animal Model

The BLUP animal model was used for BV estimation. However, there are limitations of this estimation model compared to contemporary group-based methods; we believe that since BLUP is an unbiased model and can handle fixed effects, it will give reliable results in our case. 

In the model, pedigree and database matrices were generated. The former consisted of data from full sibs, half sibs, sires, dams, and grandparents, the latter from fixed and random effects (as above) and traits. The general formula of the used BLUP animal model was as follows (Equation (4)):
*y* = *Xb* + *Za* + *e*
(4)

where “*y*” is the vector of observations; “*b*” is the vector of fixed effects; “*a*” is the vector of random animal effects; “*e*” is the vector of random residual effects; and “*X*” and “*Z*” are the incidence matrices relating records to fixed and animal effects, respectively.

Based on the guidelines of Szőke and Komlósi [44], MTDFREML software [45] was used to run the BLUP animal model for BV estimation. The used MTDFREML software automatically determined the BVs, which were copied from the result files for further calculations.

### 2.6. Estimation of Phenotypic and Genetic Correlations

The estimation of the phenotypic correlation coefficients (*r_p_*) was based on measured or scored data, whereas the estimation of genetic correlation was based on the BVs of the considered traits. Based on the instructions of Tőzsér and Komlósi [46], the following formula was used to calculate the *r_p_
*(Equation (5)):(5)$$r_{p}=\frac{\sum \left(x_{i}-\bar{x})\times (y_{i}-\bar{y}\right)}{\sqrt{\sum {\left(x_{i}-\bar{x}\right)}^{2}\times {\left(y_{i}-\bar{y}\right)}^{2}}}$$
where *r_p_* = phenotypic correlation value; *X_i_* = trait 1 data of cow; $\bar{x}$ = mean of trait 1 in entire population; *y_i_* = trait 2 data of cow; and $\bar{y}$ = mean of trait 2 in entire population.

Genetic correlation coefficients (*r_g_*) were calculated using the following formula [46] (for BV, the mean of the population was considered zero) (Equation (6)):(6)$$r_{g}=\frac{\sum \left(BV_{trait\_1})\times (BV_{trait\_2}\right)}{\sqrt{\sum {\left(BV_{trait\_1}\right)}^{2}\times {\left(BV_{trait\_2}\right)}^{2}}}$$
where *r_g_* = genetic correlation value; *BV_trait__*_1_ = breeding value of sire or animal in trait 1; and *BV_trait__*_2_ = breeding value of sire or animal in trait 2.

Both phenotypic and genetic correlation coefficients between udder conformation and production traits were calculated using SPSS 27.0 [41] software.

### 2.7. Estimation of Phenotypic and Genetic Trends

Linear regression analysis was used to estimate phenotypic and genetic trends for the traits studied. Yearly averaged data were considered according to when the cow was born and then plotted against year of birth. Phenotypic trends were based on the measured and scored results of the traits, while BVs were considered to be indicative of genetic trends. Genetic trends for sires were estimated from 1997 to 2015, while those for the whole population were estimated from 1996 to 2018. The dependent variable (Y) was the phenotypic or BV mean of the traits, the independent variable (X) was the year of birth of the cow, and the constant (a), slope (b), and goodness of fit (R^2^) values and their statistical reliability were also determined in the same way as Ostler et al. [47]. SPSS 27.0 software [41] was used for the linear regression analysis. 

The general form of the used linear regression used was as follows (Equation (7)):Y = a + bX(7)
where Y = dependent variable; a = constant; b = slope; and X = independent variable.

## 3. Results

Table 3 shows the results of the descriptive statistics of the age, udder linear scores, and production yield traits. As can be seen from the table, the production traits were quite reasonable for the breed, with 10,179.4 kg milk, 380.3 kg fat, and 333.1 kg protein in a standard lactation of 305 days [48]. The udder conformation trait scores (5.4 to 6.1) are slightly above the average of 5 on a linear scale with a total range of 1–9.

Table 4 summarizes the influences on production and udder traits estimated by REML procedure. According to the results, each of the six evaluated factors had a significant effect on the production traits. All udder conformation traits were significantly influenced by sire, herd, and year of birth. The effect of season of birth was significant on the udder conformation traits except for CL and UT, while age at evaluation was significant in all cases except for CL.

Table 5 shows the variance components and *h*^2^ estimates for the traits studied estimated by REML procedure. The latter of the production traits, MY, FY, and PY, in this study are 0.34, 0.35, and 0.30, respectively. These values are slightly higher than those published by Taylor and Field [28] (0.25) or used by Interbull [40] and Kőrösi et al. [29] (0.23, 0.26, 0.21), but fit in the range published by Guinan et al. [30] (0.21 to 0.47 for HF cows).

The *h*^2^ estimates for the evaluated udder conformation traits in our study ranged from 0.22 to 0.41, which can be described as low to medium genetic determination.

Table 6 summarizes the phenotypic and genetic correlations between production and udder conformation traits. According to the results, there is a strong phenotypic and genetic correlation (*r* = 0.61 to 0.95) between the MY traits. These results are consistent with what we knew before, that there is a strong relationship between production yield traits [28].

The phenotypic correlations (*r_p_*) between MY and udder conformation traits are weak or negligible (from −0.21 to + 0.15). The results are similar for the genetic correlation (*r_g_*), both based on the sire’s BV (from −0.33 to +0.12) and on the population base’s BV (from −0.26 to +0.15).

The association between the udder conformation traits studied was generally weak in our study, but a moderate positive correlation was observed between FU and UD (*r_p_* = 0.42, r_g_ = 0.50 or 0.57). The relationship between FU and UT showed a correlation close to the moderate values (*r_p_* = 0.36, *r_g_* = 0.33 or 0.35). These results are partly similar to and partly different from the results of Mederios et al. [16], who published genetic correlations between front and rear teat distance (0.54), UD and distance front–rear (−0.47), distance front–rear and front teat distance (0.32), and UD and front teat distance (−0.31).

Table 7 shows the results of the trend estimation udder conformation and production traits. The above typical phenotypic trend in the six udder conformation traits examined shows a slight decrease (*b* = −0.07 to −0.01), except for FTP, which shows no change (*b* = 0.00). The genetic trend is always zero to positive (*b* = 0.00 to 0.03). Both the phenotypic and genetic trends obtained in our study indicate zero or very small changes. However, these changes can be considered as practically meaningless over the ten-year period.

The data in the table show a considerable increase in MY traits over the ten-year period evaluated. The phenotypic increase (*b* = 1.6 to 42.3) is greater than the increase due to genetic effect (*b* = 0.2 to 16.5) for all three traits, although the latter is significant in all cases.

The main output of the results: The heritability of the conformation traits studied was similar to that of the milk production traits. Phenotypic and genetic correlations for the relationship between production and udder conformation were weak or negligible. Despite an increase in milk yield over the ten-year period studied, udder conformation traits did not change, but milk yield and udder conformation traits were included together in the selection index.

## 4. Discussion

The demands on the cow’s udder and the criteria for udder evaluation have changed as milk yields have increased and milking methods have evolved [3,6,7,8,9]. In the past, manual milking was less demanding on the cow’s udder. Then, with the introduction of various mechanical milking methods, the cow–human relationship and the udder requirements have also changed [6]. Therefore, the criteria for udder evaluation need to be constantly updated [3]. Furthermore, the increasing use of automatic milking, i.e., milking robots, poses new challenges to the udder conformation requirements of dairy cows and to the revision and modernization of udder judging [7,8,9].

Based on published research and practical experience, it can be concluded that the use of new milking technologies, continuous increase of MY, and changes in other production conditions may further challenge the udder requirements of cows. The technological changes and continuous increase in MY may also change the phenotypic and genetic associations of different udder traits with each other and with MY [29,49]. For these reasons, it is useful to rethink and continuously evaluate the cow’s udder conformation scoring process, genetic parameters, and changes in these traits due to selection response. All of this is necessary to adjust the methods of the conformation scoring systems of the different udder traits and to adapt them to more modern requirements and selection schemes.

It was a big challenge for us to determine how and with which method to evaluate the most important production and udder traits, their *h*^2^, phenotypic and genetic correlations, and trends. As our data came from different farms, herds, years, and seasons, the contemporary group’s method was not a viable option for us. Therefore, the BLUP animal model was used for genetic, i.e., BV estimation and the REML model for estimating variance components. From the variance components, *h*^2^, correlation, and regression of the studied production and udder traits could be evaluated.

Since all udder conformation traits in our study were significantly influenced by sire and herd, we think there is a selection opportunity to improve certain udder conformation traits. Therefore, we believe and recommend that the NAHHFB should review the udder evaluation system and strive to modernize it.

Based on the results, the *h*^2^ estimates for the evaluated udder conformation traits in our study ranged from 0.22 to 0.41, which can be described as low to medium genetic determination. Djedovic et al. [43] published a little bit lower (0.11 to 0.23) values. Our results are in line with the findings of Tribout et al. [31], who published that *h*^2^ = 0.15 to 0.45 for udder morphology traits. The results are in the range of those published by Nazar et al. [32], who reported *h*^2^ for udder traits of 0.04 to 0.49. The results in this study are also similar to those obtained by Berry et al. [33], who reported *h*^2^ estimates for the Irish HF cow population as follows: FU 0.17, RUH 0.21, udder support 0.11, UD 0.30, teat position in rear view 0.26, teat position in side view 0.19, rear teat placement 0.21, and teat length 0.34. Ismael et al. [36] reported *h*^2^ estimates for FU (0.12), FTP (0.06), front teat length (0.05), UD (0.10), RUH (0.08), and rear teat length (0.03). However, Khan and Khan [24] reported slightly higher values, according to which the *h*^2^ estimates for udder length, width, depth, and circumference were 0.68, 0.66, 0.54, and 0.60, respectively. According to Mederios et al. [16], the *h*^2^ estimates for udder balance, UD, front teat distance, rear teat distance, rear-front teat distance, and daily MY were 0.41, 0.79, 0.53, 0.40, 0.65, and 0.20, respectively.

The differences in *h*^2^ between our results and some of the literature sources mentioned may be due to the different populations and environments, and the sire and herd effects mentioned above. Considering the genetic determination of the udder conformation traits studied, as indicated by low or medium *h*^2^ values in our study, supported by similar results in the literature, it can also be declared a selection possibility. Based on these results, we believe that some udder traits in the HF breed could be further improved through strict selection to better adapt the herd to changing production and more modern milking conditions.

In our study, the negative or positive low genetic correlation between MY and the udder conformation traits studied does not indicate a meaningful association. Our results differ from the report of White and Vinson [4], who published a strong relationship between MY and udder size. Khan and Khan [24] reported moderate correlation coefficient values for the relationship between some udder traits and MY. According to their results, the *r_g_* of lactation MY with udder length, width, depth, and circumference were 0.38, 0.41, 0.35, and 0.36, respectively. Udder length had the highest *r_p_* with test day yield (0.45) in the first stage of lactation, followed by udder width (0.39), udder circumference (0.31), and UD (0.29). Tilki et al. [25] found that lactation MY was in a positive relationship with distance between anterior and posterior teats and pre-milking udder type score. Deng et al. [26] found the correlation between MY and udder circumference, length, fore UD, rear UD, and udder capacity to be 0.46, 0.64, 0.14, 016, and 0.52, respectively. Saleh et al. [27] found that the genetic correlations of udder width with total MY and persistency were strongly positive (0.86 and 0.93, respectively). However, strong negative genetic correlations were found between udder width and peak MY and lactation length (−0.92 and −0.80, respectively), rear teat length and peak MY (−0.92), and rear teat distance and persistency (−0.79).

However, most authors reported similar results to ours; according to Buchanan et al. [22], the relationship between MY and udder conformation traits was weak. Kruszyński et al. [50] found that the genetic correlation between MY and various udder traits ranged from −0.09 to +0.30 in Polish HF cows. According to Damron [23], the *r_p_* and *r_g_* values between some udder conformation traits and MY were as follows: fore udder attachment −0.09 and 0.47, RUH 0.12 and −0.13, rear udder width 0.16 and 0.09, UD −0.27 and −0.64, suspensory ligament 0.14 and 0.12, and FTP 0.02 and 0.12.

Although the results of some of the above-mentioned medium or higher correlations between udder conformation and production traits indicate that there may be an udder change with change in production, it could not be confirmed by our results and some of the literature data. We think that further studies are needed to see these relationships clearly. Our observations on the relationship between production and udder conformation, together with the literature sources, draw attention to the fact that caution must be exercised during selection aimed at improving milk yield in those udder parameters that are negatively related to production.

In our study, the association between the different udder conformation traits studied was generally weak, but a moderate positive correlation was observed between FU and UD (*r_p_* = 0.42, *r_g_* = 0.50 or 0.57). The relationship between FU and UT showed a correlation close to the moderate values (*r_p_* = 0.36, *r_g_* = 0.33 or 0.35). These results are partly similar and partly different to the results of Mederios et al. [16], who published genetic correlations between front teat distance and rear teat distance (0.54), UD and distance front–rear (−0.47), distance front–rear and front teat distance (0.32), and UD and front teat distance (−0.31).

These obtained results for correlations between udder traits, supported by the literature references, suggest that a more meaningful relationship can be expected between FU and UD. This means that if we select to improve FU, we can also improve HD, and vice versa. Therefore, we believe that some of the udder traits that are positively correlated with each other could be removed from the selection criteria and replaced with new, more important traits.

When looking at genetic trends, we observed an increase in MY traits. However, none of the udder traits studied changed (*b* = 0.00 to 0.03). Our results are partly similar and partly different to those of Theron and Mostert [34], who published increasing genetic trends, except for teat length, and a slight increasing for UD in South African HF cattle. Dube et al. [35] published that UD, rear teat placement, and FTP had a positive genetic trend, and front udder attachment and RUH had a negative genetic trend. Ismael et al. [36] found that most of the udder traits had an increasing genetic trend *(b* = 0.32 to 0.26). Exceptions were slightly negative for FUA (*b* = −0.089) and RUH (*b* = −0.062).

Based on the above, our results for the genetic trends of the udder conformation traits studied can be discussed that no changes were observed over time. Our finding that genetic selection aimed at simultaneously improving milk production, and some of the udder traits included in the selection index resulted only in an increase in milk yield, but not in changes in udder traits. This draws attention to the importance of modification of the selection procedure. Therefore, several additional aspects need to be investigated and clarified in order to further improve udder conformation in the HF cattle selection program, if there is a demand from the breeders.

## 5. Conclusions

The results, in line with several findings in the literature, showed that the genetic determination, i.e., *h*^2^ estimates of milk production traits and udder conformation traits, were similar. This would suggest that both traits are evolving as a result of simultaneous selection for both sets of traits.

However, the relationship between milk yield and udder conformation traits was found to be mostly loose. Furthermore, despite an increase in milk yield over the ten-year period studied, udder conformation traits did not change.

As the udder traits were included in the selection index, based on the results, we believe it may be necessary to reconsider the udder conformation scoring system and include a modification in the selection index.

The udder conformation traits may not only have an influence on production traits, but also on longevity and lifetime production. In our opinion, it may therefore be appropriate to examine the parameters of the udder in relation to longevity and other functional traits.

## Figures and Tables

**Table 1 animals-15-01276-t001:** Structure of the evaluated Holstein-Friesian herds in the database.

Starting Parameters	Values
Period of testing according to cow year born	2008–2018
Birth season (month) of cow	
- Winter (December + January + February)	3207
- Spring (March + April + May)	6680
- Summer (June + July + August)	3711
- Autumn (September + October + November)	1434
Number of herds	6
Number of cows	15,032
Number of sires tested (sire of cow)	666
Birth year of sires	1997–2015
Average female progeny (cow) per sire	22.57
Number of dams tested (dam of cow)	11,787
Birth date of dams	1996–2017

**Table 2 animals-15-01276-t002:** Characteristics of the used models.

Parameters	Udder Conformation Traits (Model 1)	Production Traits (Model 2)
Random effects		
- animal (cow)	+	+
Fixed effects		
- herd	+	+
- birth year of cow	+	+
- birth season of cow	+	+
- age of cow at first calving	-	+
- age of cow at scoring	+	-
Pedigree matrix		
- sire (sire of cow)	+	+
- dam (dam of cow)	+	+
- full sibs, half sibs	+	+
- grandparents	+	+
Examined traits		
- FU	+	-
- RUH	+	-
- CL	+	-
- UD	+	-
- FTP	+	-
- UT	+	-
- MY	-	+
- FY	-	+
- PY	-	+

+ = the model includes this effect; - = the model does not include this effect; FU = fore udder attachment; RUH = rear udder height; CL = central ligament; UD = udder depth; FTP = front teat placement; UT = udder texture; MY = 305-day milk yield in first lactation; FY = 305-day milk fat yield in first lactation; PY = 305-day milk protein yield in first lactation.

**Table 3 animals-15-01276-t003:** Statistical characteristics of the data for udder conformation and production traits.

Trait	Mean	SE	SD	CV%	Median	Min	Max	*p* #
AGE (month)	29.2	0.02	2.6	8.9	28.9	24.0	36.0	0.06
AFC (month)	24.8	0.02	2.0	8.1	24.5	20.0	34.0	0.07
LAC (day)	388.0	0.51	62.3	16.1	324.0	200.0	500.0	0.09
FU (score)	5.7	0.01	1.6	27.3	6.0	1.0	9.0	0.17
RUH (score)	5.4	0.01	1.1	19.5	5.0	1.0	9.0	0.21
CL (score)	6.1	0.01	1.3	21.1	6.0	1.0	9.0	0.17
UD (score)	6.1	0.01	1.4	22.5	6.0	1.0	9.0	0.19
FTP (score)	5.5	0.01	1.4	25.1	6.0	1.0	9.0	0.18
UT (score) *	5.9	0.01	1.1	18.4	6.0	2.0	9.0	0.23
MY (kg)	10,179.4	15.14	1856.6	18.2	10,216.0	5000.0	18,000.0	0.01
FY (kg)	380.3	0.55	68.0	17.9	379.7	145.8	648.5	0.01
PY (kg)	333.1	0.46	56.4	16.9	334.1	148.5	568.8	0.01

*N* = 15032; * *N* = 10502; AGE = age of cow at conformation scoring; AFC = age of cow at first calving; LAC = lactation interval; FU = fore udder attachment; RUH = rear udder height; CL = central ligament; UD = udder depth; FTP = front teat placement; UT = udder texture; MY = 305-day milk yield in first lactation; FY = 305-day milk fat yield in first lactation; PY = 305-day milk protein yield in first lactation; # Kolmogorov–Smirnov test (if *p* > 0.05, the normal distribution is confirmed.

**Table 4 animals-15-01276-t004:** Influences on udder conformation and milk yield traits.

Trait	Factors					
	Sire of Cow	Herd	Birth Year of Cow	Birth Season of Cow	Age at Scoring	Age at First Calving
Classes	666	6	11	4	4	4
FU	*p* < 0.01	*p* < 0.01	*p* < 0.01	*p* < 0.01	*p* < 0.05	-
RUH	*p* < 0.01	*p* < 0.01	*p* < 0.01	*p* < 0.05	*p* < 0.05	-
CL	*p* < 0.01	*p* < 0.01	*p* < 0.01	NS	NS	-
UD	*p* < 0.01	*p* < 0.01	*p* < 0.01	*p* < 0.01	*p* < 0.01	-
FTP	*p* < 0.01	*p* < 0.01	*p* < 0.01	*p* < 0.01	*p* < 0.01	-
UT	*p* < 0.01	*p* < 0.01	*p* < 0.05	NS	*p* < 0.01	-
MY	*p* < 0.01	*p* < 0.01	*p* < 0.01	*p* < 0.01	-	*p* < 0.01
FY	*p* < 0.01	*p* < 0.01	*p* < 0.01	*p* < 0.01	-	*p* < 0.01
PY	*p* < 0.01	*p* < 0.01	*p* < 0.01	*p* < 0.01	-	*p* < 0.01

FU = fore udder attachment; RUH = rear udder height; CL = central ligament; UD = udder depth; FTP = front teat placement; UT = udder texture; MY = 305-day milk yield in first lactation; FY = 305-day milk butterfat yield in first lactation; PY = 305-day milk protein yield in first lactation.

**Table 5 animals-15-01276-t005:** Variance components and heritability estimates for udder conformation and milk yield traits.

Traits	Parameters			
	*σ* ^2^ * _d_ *	*σ* ^2^ * _e_ *	*σ* ^2^ * _p_ *	*h*^2^*±* SE
FU	0.56	0.84	2.40	0.23 ± 0.02
RUH	0.31	0.77	1.08	0.29 ± 0.02
CL	0.34	1.24	1.58	0.22 ± 0.02
UD	0.73	1.06	1.79	0.41 ± 0.02
FTP	0.59	1.23	1.82	0.32 ± 0.02
UT	0.26	0.92	1.18	0.22 ± 0.02
MY	819,426.0	15,906,540.4	2,410,076.4	0.34 ± 0.01
FY	1163.3	2138.0	3001.3	0.35 ± 0.02
PY	655.4	1544.8	2200.2	0.30 ± 0.02

*σ*^2^*_d_* = genetic variance; *σ*^2^*_e_* = residual variance; *σ*^2^*_p_* = phenotypic variance; *h*^2^ = heritability; FU = fore udder attachment; RUH = rear udder height; CL = central ligament; UD = udder depth; FTP = front teat placement; UT = udder texture; MY = 305-day milk yield in first lactation; FY = 305-day milk butterfat yield in first lactation; PY = 305-day milk protein yield in first lactation.

**Table 6 animals-15-01276-t006:** Phenotypic and genetic correlations between udder conformation and production traits.

r	RUH	CL	UD	FTP	UT	MY	FY	PY
Phenotypic (*r_p_*)								
FU	+0.20 *	+0.09 *	+0.42 *	+0.12 *	+0.36 *	−0.04 *	−0.01	−0.03 *
RUH		+0.12 *	+0.10 *	+0.02 *	+0.17 ^&^	+0.14 *	+0.11 *	+0.13 *
CL			+0.16 *	+0.19 *	+0.10 *	+0.05 *	+0.03 *	+0.02 *
UD				+0.10 *	+0.18 *	−0.21 *	-0.17 *	−0.21 *
FTP					+0.19 *	+0.08 *	+0.09 *	+0.08 *
UT						+0.15 *	+0.11 *	+0.14 *
MY							+0.76 *	+0.95 *
FY								+0.81 *
Genetic (based on BV of sires) (*r_g_*)								
FU	+0.18 *	+0.07	+0.57 *	+0.04	+0.33 *	−0.18 *	−0.19 *	-0.18 *
RUH		+0.10 *	+0.11 *	−0.06	+0.10 *	+0.11 *	+0.10 *	+0.10 *
CL			+0.16 *	+0.09 ^&^	−0.10 *	+0.02	−0.04	−0.03
UD				+0.06	+0.17 *	−0.25 *	−0.33 *	−0.27 *
FTP					+0.19 *	+0.03	+0.08 ^&^	+0.05
UT						+0.12 *	+0.09 ^&^	+0.07
MY							+0.62 *	+0.89 *
FY								+0.68 *
Genetic (based on BV of entire population) (*r_g_*)								
FU	+0.20 *	+0.08 *	+0.50 *	+0.09 *	+0.35 *	−0.12 *	−0.12 *	−0.12 *
RUH		+0.11 *	+0.12 *	−0.02 *	+0.14 *	+0.15 *	+0.12 *	+0.13 *
CL			+0.16 *	+0.10 *	−0.08 *	+0.07 *	−0.01 ^&^	+0.02 *
UD				+0.11 *	+0.18 *	−0.21 *	−0.26 *	−0.23 *
FTP					+0.20 *	+0.01 ^&^	+0.06 *	+0.03 *
UT						+0.11 *	+0.11 *	+0.09 *
MY							+0.61 *	+0.90 *
FY								+0.69 *

* *p* < 0.01; ^&^
*p* < 0.05; BV = breeding value; FU = fore udder attachment; RUH = rear udder height; CL = central ligament; UD = udder depth; FTP = front teat placement; UT = udder texture; MY = 305-day milk yield in first lactation; FY = 305-day milk butterfat yield in first lactation; PY = 305-day milk protein yield in first lactation.

**Table 7 animals-15-01276-t007:** Results of the trend estimation for the udder conformation and production traits.

Trend	Y	Slope			Intercept			Fitting	
		*b*	SE	*p*	*a*	SE	*p*	*R* ^2^	*p*
FU									
- P	aP	−0.04	0.01	<0.01	87.44	23.54	<0.01	0.57	<0.01
- GSB	aBV	+0.02	0.00	<0.01	−33.83	8.61	<0.01	0.46	<0.01
- GAB	aBV	+0.00	0.00	<0.01	−9.56	2.69	<0.01	0.38	<0.01
RUH									
- P	aP	−0.04	0.01	<0.05	75.92	26.02	<0.05	0.45	<0.05
- GSB	aBV	+0.01	0.00	<0.01	−24.52	7.51	<0.01	0.39	<0.01
- GAB	aBV	+0.00	0.00	<0.01	−8.05	2.01	<0.01	0.44	<0.01
CL									
- P	aP	−0.00	0.01	NS	15.04	25.60	NS	0.02	NS
- GSB	aBV	−0.00	0.00	<0.10	9.13	4.67	<0.10	0.18	<0.10
- GAB	aBV	+0.00	0.00	NS	−2.00	1.46	NS	0.09	NS
UD									
- P	aP	−0.07	0.02	<0.01	155.86	34.04	<0.01	0.68	<0.01
- GSB	aBV	+0.03	0.01	<0.01	−56.73	9.82	<0.01	0.64	<0.01
- GAB	aBV	+0.01	0.00	<0.01	−19.39	4.30	<0.01	0.49	<0.01
FTP									
- P	aP	+0.00	0.01	NS	2.61	24.59	NS	0.00	NS
- GSB	aBV	+0.02	0.01	<0.01	−38.54	11.71	<0.01	0.38	<0.01
- GAB	aBV	+0.01	0.00	<0.01	−17.66	3.15	<0.01	0.60	<0.01
UT									
- P	aP	−0.02	0.01	NS	55.66	37.21	NS	0.23	NS
- GSB	aBV	+0.01	0.00	<0.01	−23.45	4.94	<0.01	0.54	<0.01
- GAB	aBV	+0.00	0.00	<0.01	−5.70	1.20	<0.01	0.53	<0.01
MY									
- P	aP	+42.3	24.8	NS	−74,870.7	49,850.5	NS	0.25	NS
- GSB	aBV	+16.5	6.2	<0.05	−32,974.8	12,517.8	<0.05	0.29	<0.05
- GAB	aBV	+5.5	2.3	<0.05	−10,968.5	4522.3	<0.05	0.24	<0.05
FY									
- P	aP	+2.2	0.6	<0.01	−3993.3	1161.8	<0.01	0.61	<0.01
- GSB	aBV	+0.5	0.3	<0.10	−989.9	526.1	<0.10	0.18	<0.10
- GAB	aBV	+0.3	0.1	<0.01	−530.2	168.7	<0.01	0.32	<0.01
PY									
- P	aP	+1.6	0.8	<0.10	−2980.9	1696.0	NS	0.30	<0.10
- GSB	aBV	+0.5	0.2	<0.05	−986.6	401.5	<0.05	0.27	<0.05
- GAB	aBV	+0.2	0.1	<0.01	−400.8	114.2	<0.01	0.39	<0.01

NS = not significant; FU = fore udder attachment; RUH = rear udder height; CL = central ligament; UD = udder depth; FTP = front teat placement; UT = udder texture; MY = 305-day milk yield in first lactation; FY = 305-day milk butterfat yield in first lactation; PY = 305-day milk protein yield in first lactation; P = phenotypic trend; GSB = genetic trend based on BV of sires; GAB = genetic trend based on BV of entire population; aP = average phenotypic data of the trait; aBV = average BV.

## Data Availability

The data presented in this study are available on request from the National Association of Hungarian Holstein Friesian Breeders.

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
