# Peer review of "Heritability and Trends in Selected Udder Traits and Their Relation to Milk Production in Holstein-Friesian Cows"

_animals, 2025, doi:10.3390/ani15091276_

Round 1
Reviewer 1 Report (Previous Reviewer 3)
Comments and Suggestions for Authors
Estimating the heritability (h²) of important udder conformation traits, their interrelationships, and associations with production traits, as well as analyzing udder conformation and genetic trends in Holstein Friesian cows, is essential for improving selection procedures in high-yielding breeds. In the revised manuscript, the authors included two different models for the estimation of variance components and population genetic parameters using the REML method. However, I could not find any significant differences in their results for variance components and population genetic parameters compared to the previous version of the manuscript. Still, I have the following unanswered questions that I would like to clarify:
First, including too many udder traits may complicate heritability estimation due to potential redundancy and collinearity. A proper justification for evaluating all traits separately is needed, or alternatively, a composite trait approach should be considered to provide a clearer picture of heritability variability.
Second, the study spans a long period (1996–2017), during which seasonal variations could have significantly impacted milk yield and udder health, potentially affecting heritability estimates. However, the authors did not account for seasonal effects (except birth season of cow), which may introduce confounding factors. A more comprehensive analysis incorporating seasonal influences would strengthen the reliability of the findings, or at the very least, the authors should provide a justification for omitting this variable.
Author Response
Response to the Reviewer 1
Dear Reviewer,
Thank you for recognising that estimating the heritability (h²) of important udder conformation traits, their interrelationships and associations with production traits, and analysing udder conformation and genetic trends in Holstein-Friesian cows is essential for improving selection procedures in high yielding breeds.
Comment: Estimating the heritability (h²) of important udder conformation traits, their interrelationships, and associations with production traits, as well as analyzing udder conformation and genetic trends in Holstein Friesian cows, is essential for improving selection procedures in high-yielding breeds. In the revised manuscript, the authors included two different models for the estimation of variance components and population genetic parameters using the REML method. However, I could not find any significant differences in their results for variance components and population genetic parameters compared to the previous version of the manuscript. Still, I have the following unanswered questions that I would like to clarify:
Response: We used two different models because one of the reviewers suggested that we try a different model as the results differed depending on the model. The identical results confirmed our previous experience.
Comment: First, including too many udder traits may complicate heritability estimation due to potential redundancy and collinearity. A proper justification for evaluating all traits separately is needed, or alternatively, a composite trait approach should be considered to provide a clearer picture of heritability variability.
Response: Thank you for bringing this to our attention. We have studied the different udder traits separately. We agree that it would have been more appropriate to evaluate the udder composite index. However, we wanted to look at each udder trait separately to see which ones should be removed and replaced with others, or new udder traits added to the composite for the breeder's decision.
Comment: Second, the study spans a long period (1996–2017), during which seasonal variations could have significantly impacted milk yield and udder health, potentially affecting heritability estimates. However, the authors did not account for seasonal effects (except birth season of cow), which may introduce confounding factors. A more comprehensive analysis incorporating seasonal influences would strengthen the reliability of the findings, or at the very least, the authors should provide a justification for omitting this variable.
Response: We agree that seasonal effects are very important as they influence both milk production and udder health. Several seasonal variations could have been evaluated (e.g. first calving season, first calving season, etc.). However, as the birth season influences the seasons of the other events mentioned, we felt it was appropriate to look at this one. As lactation lasts ten months or more, it covers several seasons. Therefore, the effects of the different seasons cancel each other out and do not influence the final result.

Reviewer 2 Report (New Reviewer)
Comments and Suggestions for Authors
The paper is typical for such study. Heritability calculation is based on the large amount of data. There is no comments about lack of normal dustribution for udder scores. Age is typical quantitative traits and making groups is not needed - better use this trait as independent in the model.
Few details:
in formula 5 and 6 in the denominator the second "sum sign" is needed - it should be sum of sguares times sum of squares
in lines 115-117 should be 10 traits but there are only 9 and they should better axplain in this place
In table 3 SE are not equal to zero so better show tham les some values
Author Response
Response to the Reviewer 2
Dear Reviewer,
Thank you very much for taking the time and energy to review our manuscript. We provide the following responses to your suggestions:
Comment: The paper is typical for such study. Heritability calculation is based on the large amount of data. There are no comments about lack of normal distribution for udder scores. Age is typical quantitative traits and making groups is not needed - better use this trait as independent in the model.
Response: Thank you for your appreciation that the paper is typical for such study.
Comment: in formula 5 and 6 in the denominator the second "sum sign" is needed - it should be sum of squares times sum of squares
Response: We apologize, the denominator in formula 5 was incorrect. We have corrected it. After reviewing it several times, both formulas 5 and 6 now appear to be correct.
Comment: in lines 115-117 should be 10 traits but there are only 9 and they should better axplain in this place
Response: In Hungary, in the current scoring system, 21 traits are evaluated, of which 10 traits relate to the udder. During our work, we thoroughly reviewed these 10 udder traits, and with the help of the Holstein association's staff, we selected the 6 most important of them. We added 3 milk yield traits to this, so we evaluated a total of 9 traits during our work.
To clarify this, we have modified the materials and methods section as follows: “Of the 10 udder traits introduced earlier the reason why we have chosen the previous 6 mentioned udder traits is that, according to several literary sources [4,16,22,23,24,25,26], they have the greatest influence on milk production.” (line 148-151)
Comment: In table 3 SE are not equal to zero so better show them les some values
Response: The SE values in Table 3 were rounded, but because they were too small, their value rounded to one decimal place became zero. In the table, SE values have been adjusted to two decimal places.

Reviewer 3 Report (New Reviewer)
Comments and Suggestions for Authors
The authors conducted a study aimed at investigating phenotypic and genetic trends in udder traits and their relationship with milk productivity in cows.
The title of the paper is descriptive and the abstract corresponds to the main text. The introduction provides a good overview of the issue being studied and justifies the relevance of the research conducted.
The study was conducted on a large sample, and the data collected reflect a long period of time. The statistical methods and models used for the analysis are described in detail.
The results are described quite clearly and supported by informative tables.
I would like to recommend this manuscript for publication in Animals.
The authors have done a great job, but there are a few questions just for clarification:
Were the same bulls used in different herds? Or were some bulls used in some herds and other bulls used in other herds? Were the bulls closely related?
Author Response
Response to the Reviewer 3
Dear Reviewer,
Thank you very much for taking the time and energy to review our manuscript. We provide the following responses to your suggestions:
Comment: The authors conducted a study aimed at investigating phenotypic and genetic trends in udder traits and their relationship with milk productivity in cows. The title of the paper is descriptive and the abstract corresponds to the main text. The introduction provides a good overview of the issue being studied and justifies the relevance of the research conducted. The study was conducted on a large sample, and the data collected reflect a long period of time. The statistical methods and models used for the analysis are described in detail. The results are described quite clearly and supported by informative tables. I would like to recommend this manuscript for publication in Animals.
Response: Thank you for your appreciation of our study, that the authors have done a great job.
Comment: The authors have done a great job, but there are a few questions just for clarification: Were the same bulls used in different herds? Or were some bulls used in some herds and other bulls used in other herds? Were the bulls closely related?
Response: In the studied 6 herds the same bulls were used. The ratio of their progeny in different herds was similar to some extent.

Reviewer 4 Report (New Reviewer)
Comments and Suggestions for Authors
In this manuscript, the authors describe the heritability and related trends in milk production and udder traits. The study is well-designed and the results are presented well. However, the manuscript needs a major revision for publication. Specific comments are given below.
Please give an overview of the association of udder traits to production parameters. How udder traits influence the production traits? Provide the biological mechanisms. Please support with appropriate references.
Introduction
Line 47-from 1800s -in ?
Line 61-62-There are publications reporting an association between udder morphology and longevity in dairy cows [4,5,17,18,19,20,21]. Good point. Please elaborate explaining the possible mechanisms, with appropriate references.
Line 64- There is an association between some udder conformation traits and milk yield in cows. -Okay. The association may be causal or otherwise. Statements need supporting data. Please explain the mechanism with appropriate references.
Line 120-123. Based on the discrepancy cited above, this study aimed to evaluate the h2 estimates 120 of some important udder conformation traits (FU, RUH, CL, UD, FTP, and UT), their re- 121 relationship with each other and with production yield, as well as their phenotypic and 122 genetic trends over a 10-year period in HF cows within a relatively high milk yield level. Please explain with appropriate references, why these six traits were selected out of 21 (or more) udder traits?
Line 123- 10-year period -please change to 10 years
Give an overview of other non-genetic factors affecting udder traits.
Materials and methods
Give an overview of the non-genetic factors affecting udder traits like breed, nutrition, and milking methods? or production systems. Explain briefly how those confounding factors were controlled in the study.
How did you record udder traits? I would add a picture to show how udder traits were measured.
Results
Is it possible to derive some useful output such as the most significant udder trait you found in the study associated with production traits? I would give some graphical/pictorial representation.
Discussion
Please discuss the results. The discussion is mostly assumptions based on the results. Please explain the underlying phenomena with appropriate references.
Line 325-333-Good points. But all are assumptions without supportive references. Please add the mechanism and give appropriate references.
Comments on the Quality of English LanguageMust be improved
Author Response
Response to the Reviewer 4
Dear Reviewer,
Thank you for your appreciation that the study is well-designed and the results are presented well.
Comment: In this manuscript, the authors describe the heritability and related trends in milk production and udder traits. The study is well-designed and the results are presented well. However, the manuscript needs a major revision for publication. Specific comments are given below. Please give an overview of the association of udder traits to production parameters. How udder traits influence the production traits? Provide the biological mechanisms. Please support with appropriate references.
Response: This paper does not deal with the physiology and biology of the udder and milk production, but with the breeding aspect. For this reason, and for reasons of space, we did not wish to go into detail on the proposed biological correlations. We have only added the following section to the manuscript: "Milk secretion and milk yield depend on udder elasticity, blood supply and glandular composition, which are related to udder size and conformation [4,22]. The same udder diseases are also related to milk production [4,22,23]. Deeper udders are associated with an increased incidence of mastitis, resulting in lower milk yield. Higher yielding cows tend to have larger udders [4,23,24]". (line 65-70)
Introduction
Comment: Line 47-from 1800s -in ?
Response: It is corrected to in 1800s. (line 45)
Comment: Line 61-62: There are publications reporting an association between udder morphology and longevity in dairy cows [4,5,17,18,19,20,21]. Good point. Please elaborate explaining the possible mechanisms, with appropriate references.
Response: The following sentences have been added: "The explanation for this is that if the udder develops unfavorably, the cow's milk production is lower and the risk of mastitis is higher, so the cows are culled earlier and their productive life is shorter.” (line 60-63)
Comment: Line 64: There is an association between some udder conformation traits and milk yield in cows. -Okay. The association may be causal or otherwise. Statements need supporting data. Please explain the mechanism with appropriate references.
Response: The explanation is the same as in comment regarding in line 66-71. The sentence “There is an association between some udder conformation traits and milk yield in cows” has been completed as follows: “There is an association between some udder conformation traits and milk yield in cows due to some biological mechanisms [4,22,23,24]. (line 64-65)
Comment: Line 120-123. Based on the discrepancy cited above, this study aimed to evaluate the h2 estimates 120 of some important udder conformation traits (FU, RUH, CL, UD, FTP, and UT), their re- 121 relationship with each other and with production yield, as well as their phenotypic and 122 genetic trends over a 10-year period in HF cows within a relatively high milk yield level. Please explain with appropriate references, why these six traits were selected out of 21 (or more) udder traits?
Response: Among the linear traits defined and recommended for all countries by the Type Classification Working Group of the World Holstein Friesian Federation, the udder-related traits provided suitable baseline data for conducting the analysis during the study period.
The following sentence has been added: "Of the 10 udder traits introduced earlier the reason why we have chosen the previous 6 mentioned udder traits is that, according to several literary sources [4,16,22,23,24,25,26], they have the greatest influence on milk production.” (line: 148-151)
Comment: Line 123: 10-year period -please change to 10 years
Response: It has been replaced to 10 years. (line 129)
Materials and methods
Comment: Give an overview of the non-genetic factors affecting udder traits like breed, nutrition, and milking methods? or production systems. Explain briefly how those confounding factors were controlled in the study.
Response: The statement was added as follows. "There are several non-genetic factors such as nutrition and milking methods or production systems that influence udder characteristics and production. These effects were included in the herd effect, which was considered as a fixed effect in the model used.” (line 184-187)
Comment: How did you record udder traits? I would add a picture to show how udder traits were measured.
Response: It is written in line 151-152: “Scoring was performed by the same association specialist on a linear 1-9 udder score scale [2,3].”
Linear conformation scoring is conducted in accordance with the definitions and reference scale established by the World Holstein Friesian Federation (WHFF), adapted to the trait-specific population distribution curves of each member country. Member nations calibrate the midpoint and extreme values of individual traits to reflect the normal distribution within their national herds, and these adjusted values are applied during evaluations. Trait assessment is performed solely through visual appraisal, without the use of measuring instruments. Evaluations are carried out following standardized principles agreed upon during international (WHFF) and European (EHRC) classification workshops. In Hungary, harmonization is maintained through four annual calibration sessions for conformation classifiers, during which statistical analyses are used to monitor and assess scoring trends for each trait.
There are pictures in the cited literature. This is 6 times 9, or a total of 54 images, which we have not included in the manuscript due to space limitations.
Results
Comment: Is it possible to derive some useful output such as the most significant udder trait you found in the study associated with production traits? I would give some graphical/pictorial representation.
Response: As expected, the research confirmed that fore udder attachment, udder depth, rear udder width, and udder texture were the most influential linear traits in relation to milk production.
The following sentences have been added at the end of the results chapter: "The main output of the results: The heritability of the conformation traits studied was similar to that of the milk production traits. Phenotypic and genetic correlations for the relationship between production and udder conformation were weak or negligible. Despite an increase in milk yield over the ten years period studied, udder conformation traits did not change, but milk yield and udder conformation traits were included together in the selection index.” (line 326-331).
Discussion
Comment: Please discuss the results. The discussion is mostly assumptions based on the results. Please explain the underlying phenomena with appropriate references.
Response: The conclusions section has been modified as above.
Comment: Line 325-333-Good points. But all are assumptions without supportive references. Please add the mechanism and give appropriate references.
Response: As the transition to robotic milking becomes increasingly widespread, the importance of udder conformation traits is gaining more recognition. In the near future, particular attention will be paid to udder depth, the placement of the front and rear teats, the attachment of the fore udder, and the quality of udder suspension.
A previously cited [29] and a new reference [49] has been inserted into the manuscript. The new reference is as follows (line 345 and line 584-585):
Bognár, L.; Szabó, F. Management of “modern” Holstein cows focusing on sustainability and resilience – Review of recent achievements. Chem. Eng. Trans. 2023, 107, 169–174. https://doi.org/10.3303/CET23107029

Round 2
Reviewer 4 Report (New Reviewer)
Comments and Suggestions for Authors
The authors have modified the manuscript according to the suggestions. Hence, I recommend publication.
This manuscript is a resubmission of an earlier submission. The following is a list of the peer review reports and author responses from that submission.
Round 1
Reviewer 1 Report
Comments and Suggestions for Authors
In this study, the authors computed heritability estimates for several udder conformation traits in Holstein-Friesian milk cattle from Hungary for a period of ten years. Phenotypic correlation with milk yield was also assessed. The authors note that although milk yield increased during the period of study, udder conformation traits were virtually unchanged. As such, the authors propose the reconfiguration of the udder scoring system and its possible removal from the selection index.
The population from this study consists of 15,032 animals from 6 herds. This population size is adequate for this type of study and is expected to yield good estimates for genetic parameters of phenotypes of interest. The single-step BLUP animal model was used for computing genetic parameters, implemented using the MTDFREML. This methodology and software are quite standard in the field and good choices for this study.
The results are well presented and compared to the preexisting literature, and the authors’ conclusions logically follow from the results that were obtained. Overall, while the results aren’t quite groundbreaking and do not present something entirely novel, the article may still attract a moderate dose of interest from readers, especially within the field of dairy milk farming. The suggestions for revising the udder scoring system and the selection index in Hungary also have the potential to translate to impactful changes at a national level.
Some specific remarks:
L49-L50: “the conformation related to milk production, its favorable udder is very important” – this phrasing is quite strange and difficult to understand. I assume the authors meant to write that “proper udder conformation is important for milk production”
L67: “published” -> “found”
L114: “genetic trends” -> “genetic trend”
L114: “and slightly negative for RUH” – this needs rephrasing as it does not fit well with the rest of the phrase
L117-L120: “the traditional udder scoring practice will retain its importance in the future” – this should be a separate sentence, or it should be better linked with the beginning of the phrase
L125: “teeth placement” – did the authors mean “teat placement”?
L154: the name “Used database” for the second column of Table 1 seems wrong. Something like “Values” would be better. Also, “Birth date of sires” should be replaced with “Birth year of sires”
L157: the authors should specify which version of the MTDFREML software was used
L177: the authors should specify which software package was used to implement the linear regression analysis
L188: “the production traits were quite reasonable for the breed” – the authors should also include references for the average production traits of the breed
L202: “was significant except for CL” -> “was significant in all cases except for CL”
L236: the title la Table 6 is wrong and doesn’t reflect its contents
L278: “results obtained” -> “obtained results”
L280: “the udder conformation traits tested” -> “the tested udder conformation traits”
L282: “trend calculation” -> “trend estimation”
L292: “increasing” -> “increase”
L318-L320: in addition to the suggestions for revising the udder conformation scoring system and the selection index, the authors should also include some suggestions for future work, which could improve or continue the work and the results from this study
The manuscript is generally well written, although it suffers from minor issues that should be addressed. The use of English, with some exceptions, did not prevent the understanding of the article. However, small issues for grammar, spelling and punctuation are found throughout the manuscript and a full revision of the text for correcting these mistakes will be required.
Author Response
Response to the Reviewer 1
Dear Reviewer, Thank you for your careful review of our work and your comments. We have corrected the issues raised, which we hope will contribute to making our paper suitable for publication.
Comment: In this study, the authors computed heritability estimates for several udder conformation traits in Holstein-Friesian milk cattle from Hungary for a period of ten years. Phenotypic correlation with milk yield was also assessed. The authors note that although milk yield increased during the period of study, udder conformation traits were virtually unchanged. As such, the authors propose the reconfiguration of the udder scoring system and its possible removal from the selection index.
The population from this study consists of 15,032 animals from 6 herds. This population size is adequate for this type of study and is expected to yield good estimates for genetic parameters of phenotypes of interest. The single-step BLUP animal model was used for computing genetic parameters, implemented using the MTDFREML. This methodology and software are quite standard in the field and good choices for this study.
The results are well presented and compared to the pre-existing literature, and the authors’ conclusions logically follow from the results that were obtained. Overall, while the results aren’t quite groundbreaking and do not present something entirely novel, the article may still attract a moderate dose of interest from readers, especially within the field of dairy milk farming. The suggestions for revising the udder scoring system and the selection index in Hungary also have the potential to translate to impactful changes at a national level.
Response: Thank you. In our work, we attempted to establish the relationship between udder morphology and milk production using a large database and a modern BLUP model. We hope that these results will be of interest to Holstein-Friesian breeders not only in Hungary, but also in any part of the world.
Some specific remarks:
Comment: L49-L50: “the conformation related to milk production, its favorable udder is very important” – this phrasing is quite strange and difficult to understand. I assume the authors meant to write that “proper udder conformation is important for milk production”
Response: Thank you very much for your comment, the English corrections are considered very useful. The sentence was changed. (line 51)
Comment: L67: “published” -> “found”
Response: The beginning of the sentence has been changed. (line 66)
Comment: L114: “genetic trends” -> “genetic trend”
Response: Thank you for Your English language comment, we have corrected it. (line 98)
Comment: L114: “and slightly negative for RUH” – this needs rephrasing as it does not fit well with the rest of the phrase
Response: The sentence has been simplified as follows: “Ismael et al. [36] have found that the most of the udder traits have an increasing genetic trend, except the fore udder attachment (FU) and RUH.” (line 97-99)
Comment: L117-L120: “the traditional udder scoring practice will retain its importance in the future” – this should be a separate sentence, or it should be better linked with the beginning of the phrase
Response: The said sentence was divided into two parts. As suggested, the second part of the sentence was added to the end of the first sentence of the paragraph. The corrected paragraph became as follows: “In view of the above, morphological evaluation and scoring of cows' udders is essential [18,19,32,37], the traditional udder scoring practice will retain its importance in the future [31]. However, there are modern genome-wide association studies for udder traits which suggest that many candidate genes could provide valuable information on the genetic architecture of udder traits [38,39].” (line 100-104)
Comment: L125: “teeth placement” – did the authors mean “teat placement”?
Response: Yes. We apologize for the typo. (line 112)
Comment: L154: the name “Used database” for the second column of Table 1 seems wrong. Something like “Values” would be better. Also, “Birth date of sires” should be replaced with “Birth year of sires”
Response: The terms in the header and first column of the Table 1 has been corrected as suggested. (line 125)
Comment: L157: the authors should specify which version of the MTDFREML software was used
Response: The issue of the version number of the MTDFREML software was already discussed in connection with a previous paper (https://doi.org/10.3390/ani14172513). We were unable to provide the exact version number of the software we used, so the publisher provided the following link to the software: https://zzlab.net/MTDFREML/index.html#
We have included this link in this manuscript as well. (line 469-470)
Comment: L177: the authors should specify which software package was used to implement the linear regression analysis
Response: SPSS software was used for linear regression analysis. We have added a sentence at the end of the said paragraph: “SPSS 27.0 [42] software was used for linear regression analysis.” (line 167-168)
Comment: L188: “the production traits were quite reasonable for the breed” – the authors should also include references for the average production traits of the breed
Response: The reference can be that in Hungary in 2024 year the average production of 133.846 Holstein cows was 10.842 kg milk, 415,9 kg butterfat and a 366,6 kg milk protein. It is inserted in the manuscript. (line 174-176)
Comment: L202: “was significant except for CL” -> “was significant in all cases except for CL”
Response: Thank you very much for the comment on English grammar, we the sentence was corrected. (line 188)
Comment: L236: the title la Table 6 is wrong and doesn’t reflect its contents
Response: We apologize for the inaccuracy and incorrect title of Table 6. The title of Table 6 has been changed to: “Phenotypic and genetic correlations between udder conformation and production traits in Holstein-Friesian cows.” (line 208-209)
Comment: L278: “results obtained” -> “obtained results”
Response: Thank you, corrected. (line 333)
Comment: L280: “the udder conformation traits tested” -> “the tested udder conformation traits”
Response: We rephrased this sentence in the manuscript, and did not use the grammatical structure to be corrected.
Comment: L282: “trend calculation” -> “trend estimation”
Response: Thank you, corrected. (line 228)
Comment: L292: “increasing” -> “increase”
Response: Thank you, corrected. (line 237)
Comment: L318-L320: in addition to the suggestions for revising the udder conformation scoring system and the selection index, the authors should also include some suggestions for future work, which could improve or continue the work and the results from this study
Response: We have edited the following paragraph at the end of the conclusions section: “The udder conformation traits may not only have an influence on production traits, but also on longevity and lifetime production. In our opinion, it may therefore be appropriate to examine the parameters of the udder in relation to longevity.” (line 361-363)
Comments on the Quality of English Language
Comment: The manuscript is generally well written, although it suffers from minor issues that should be addressed. The use of English, with some exceptions, did not prevent the understanding of the article. However, small issues for grammar, spelling and punctuation are found throughout the manuscript and a full revision of the text for correcting these mistakes will be required.
Response: Thank you very much for the numerous comments and suggestions related to English usage and grammar. We trust that with these corrections the English language of the manuscript has been greatly improved. In addition, we have improved and clarified the English language in numerous places in the manuscript. We hope that with these changes, the "Englishness" of the manuscript has become appropriate and acceptable.

Reviewer 2 Report
Comments and Suggestions for Authors
Dear all,
The manuscript “Heritability and trends in selected udder traits and their relation to milk production in Holstein-Friesian cows” had the goal of “to evaluate the heritability estimates of some important udder conformation traits, their relationship to each other and with production, and their phenotypic and genetic trends over a 10-year period in relatively high yielding Holstein Friesian cows”. After a thorough evaluation, I recommend rejecting the manuscript due to significant structural and content issues that compromise its scientific value.
1. Introduction: The introduction is excessively long, resembling a literature review rather than a concise presentation of the study's background. It includes points that are more appropriate for a discussion section but fails to clearly articulate the research problem the study aims to address.
2. Materials and Methods: The methodology lacks critical details, particularly regarding the assumptions underlying the model used. Without a clear description of these assumptions, it is impossible to assess the validity and applicability of the study’s approach.
3. Results and Discussion: The discussion section does not provide a meaningful interpretation of the study’s findings. Instead of critically analyzing the results, it consists mainly of unrelated references to other studies, without integrating them into a coherent discussion relevant to the data presented. There is no substantial discussion of the study’s findings, their implications, or their limitations.
Given these fundamental issues, the manuscript does not meet the necessary standards for publication. A major revision would be required to address these deficiencies, including restructuring the introduction, providing a more transparent methodology, and ensuring that the discussion focuses on the study’s findings rather than unrelated scientific commentary.
Author Response
Response to the Reviewer 2
We are grateful for the reviewer's comments, which help us greatly to improve our manuscript. Our point-by-point response is given below:
Comment: 1. Introduction: The introduction is excessively long, resembling a literature review rather than a concise presentation of the study's background. It includes points that are more appropriate for a discussion section but fails to clearly articulate the research problem the study aims to address.
Response: We accept your comment that the introduction chapter is too long and includes a literature review.
Based on your comments, we have shortened the introduction and moved several references to the discussion chapter. Hope this change gave better highlight the background to the research.
Comment: 2. Materials and Methods: The methodology lacks critical details, particularly regarding the assumptions underlying the model used. Without a clear description of these assumptions, it is impossible to assess the validity and applicability of the study’s approach.
Response: We agree that the methodology lacks critical details, particularly regarding the assumptions underlying the model used. To be honest, we did not want to make assumptions in the manuscript about which models are appropriate for the purpose of the study and which are not.
We have not even mentioned the methods that could not be considered. The methods that were appropriate for the purpose of the study were presented. The methods used are up to date and reliable based on the literature references.
However, there have been a number of changes to the methodology section as a result of your comments. For example:
The issue of the version number of the MTDFREML software was already discussed in connection with a previous paper (https://doi.org/10.3390/ani14172513). We were unable to provide the exact version number of the software we used, so the publisher provided the following link to the software: https://zzlab.net/MTDFREML/index.html# (line 469-470)
SPSS software was used for linear regression analysis. We have added a sentence at the end of the said paragraph: “SPSS 27.0 [42] software was used for linear regression analysis.” (line 167-168)
Comment: 3. Results and Discussion: The discussion section does not provide a meaningful interpretation of the study’s findings. Instead of critically analyzing the results, it consists mainly of unrelated references to other studies, without integrating them into a coherent discussion relevant to the data presented. There is no substantial discussion of the study’s findings, their implications, or their limitations.
Response: The Results and Discussion sections have been split into chapters. Results, Discussion appear as separate chapters in the revised manuscript. We have tried to critically analyse the results and emphasise the substantive discussion of the study findings, their implications and also their limitations.
Comment: Given these fundamental issues, the manuscript does not meet the necessary standards for publication. A major revision would be required to address these deficiencies, including restructuring the introduction, providing a more transparent methodology, and ensuring that the discussion focuses on the study’s findings rather than unrelated scientific commentary.
Response: Thank you for your summarised comments. We have tried to make the major revision by restructuring the introduction, completing the methodology and improving the discussion section.

Reviewer 3 Report
Comments and Suggestions for Authors
Estimating the heritability (h²) of important udder conformation traits, their interrelationships, and associations with production traits, as well as analyzing their phenotypic and genetic trends in Holstein Friesian cows, is essential for improving selection procedures in high-yielding breeds. The authors evaluated six udder conformation traits—fore udder attachment (FU), rear udder height (RUH), central ligament (CL), udder depth (UD), front teat placement (FTP), and udder texture (UT)—using a 1–9 linear udder score scale. However, two key concerns arise.
First, including too many udder traits may complicate heritability estimation due to potential redundancy and collinearity. A proper justification for evaluating all traits separately is needed, or alternatively, a composite trait approach should be considered to provide a clearer picture of heritability variability. Additionally, the subjective scoring system could introduce observer bias, particularly for udder texture (UT), which is challenging to quantify.
Second, the study spans a long period (1996–2017), during which seasonal variations could have significantly impacted milk yield and udder health, potentially affecting heritability estimates. However, the authors did not account for seasonal effects, which may introduce confounding factors. A more comprehensive analysis incorporating seasonal influences would strengthen the reliability of the findings, or at the very least, the authors should provide a justification for omitting this variable.
Author Response
Response to the Reviewer 3
Thank you for your helpful comments and corrections. We have tried to improve the manuscript according to your suggestions.
Comment: Estimating the heritability (h²) of important udder conformation traits, their interrelationships, and associations with production traits, as well as analyzing their phenotypic and genetic trends in Holstein Friesian cows, is essential for improving selection procedures in high-yielding breeds. The authors evaluated six udder conformation traits - fore udder attachment (FU), rear udder height (RUH), central ligament (CL), udder depth (UD), front teat placement (FTP), and udder texture (UT) - using a 1–9 linear udder score scale. However, two key concerns arise. First, including too many udder traits may complicate heritability estimation due to potential redundancy and collinearity. A proper justification for evaluating all traits separately is needed, or alternatively, a composite trait approach should be considered to provide a clearer picture of heritability variability. Additionally, the subjective scoring system could introduce observer bias, particularly for udder texture (UT), which is challenging to quantify.
Response: We agree that including too many udder traits may complicate heritability estimation due to potential redundancy and collinearity. Therefore, in our study we performed heritability estimates for each trait separately.
Thank you for your suggestion that a composite trait should be considered to give a clearer picture of heritability variability. In the next part of our study, based on these results, we would like to look at composite traits.
Agree that the subjective scoring system could introduce observer bias, especially for udder texture (UT), which is difficult to quantify. However, we believe that a large number of observations can statistically reduce the error.
Comment: Second, the study spans a long period (1996-2017), during which seasonal variations could have significantly impacted milk yield and udder health, potentially affecting heritability estimates. However, the authors did not account for seasonal effects, which may introduce confounding factors. A more comprehensive analysis incorporating seasonal influences would strengthen the reliability of the findings, or at the very least, the authors should provide a justification for omitting this variable.
Response: Agree that seasonal variation could have had a significant effect on milk yield and udder health, potentially affecting heritability estimates.
It should be noted that we have evaluated seasonal effects. Both cow year of birth and season of birth were included as fixed effects in the model used (Table 2). According to the results, both effects had a significant influence on milk production traits, birth year for each udder trait and season of birth on FU, RUH, UD and FTP (Table 4). We hope that the model we used gave acceptable estimates of heritability.

Round 2
Reviewer 2 Report
Comments and Suggestions for Authors
Dear all,
Thank you for providing the revised version of the manuscript “Heritability and trends in selected udder traits and their relation to milk production in Holstein-Friesian cows.” I appreciate the effort made to address the concerns raised in the initial review.
However, after a thorough evaluation, I believe that significant structural and content issues remain, which compromise the scientific value of the study.
The Introduction section, while improved, still does not fully meet scientific writing standards. It contains single sentences formatted as paragraphs, includes irrelevant discussions where a proper introduction should be provided, and lacks a clear presentation of the importance of the research topic. Additionally, some statements are presented as definitive facts without proper support.
Regarding the Materials and Methods section, one concern I have is the treatment of fixed effects. Specifically, the year, herd, and season were included as separate effects in the model rather than as a contemporaneous group, which would better capture the interaction between these factors. Additionally, the definition of "season" should be clarified, and the total number of observations for each analyzed trait should be provided. The section on variance components and genetic parameter estimation lacks details about the method used for estimation. Furthermore, were all animals evaluated at the same parity number, and were they primiparous or multiparous?
The Results and Discussion sections have not undergone significant revisions, leaving the implications and limitations of the study unclear.
At this stage, I am unable to endorse the manuscript for publication, as it does not meet the journal's standards.
Author Response
Response to the Reviewer 2 (round 2)
Dear Reviewer,
Thank you for your appreciation of our effort to make our manuscript better. Also, thank for your further comments that can help us to make further improvements.
Comment: After a thorough evaluation, I believe that significant structural and content issues remain, which compromise the scientific value of the study.
Response: Thank you for your thorough evaluation and for drawing our attention to significant structural and content issues that compromise the scientific value of the study. We have made further structural and content changes in our manuscript.
Comment: The Introduction section, while improved, still does not fully meet scientific writing standards. It contains single sentences formatted as paragraphs, includes irrelevant discussions where a proper introduction should be provided, and lacks a clear presentation of the importance of the research topic. Additionally, some statements are presented as definitive facts without proper support.
Response: We have tried to combine single sentences into a common paragraph format, reframe irrelevant discussions, provide a proper introduction to each, and more clearly present the importance of the topic. Furthermore, we have tried to adequately support each statement. For this reason, the introduction has been reworded in several places. Some new sentences have also been added.
Comment: Regarding the Materials and Methods section, one concern I have is the treatment of fixed effects. Specifically, the year, herd, and season were included as separate effects in the model rather than as a contemporaneous group, which would better capture the interaction between these factors.
Response: We agree that creating contemporaneous groups would better capture the interaction between the factors being assessed. However, in our work, all fixed effects were treated separately to see their separate effects. This study was a follow-up to our previously published paper (https://doi.org/10.3390/ani14182753). So we did not want to change the method for better comparability. In our next planned study we would like to evaluate the interactions between different factors on different udder traits.
Comment: Additionally, the definition of "season" should be clarified, and the total number of observations for each analyzed trait should be provided.
Response: Cows were classified into birth seasons based on their birth dates. If the cow was born in December, January or February, then the birth season was winter. Along the same logic: March + April + May = spring; June + July + August = summer; September + October + November = autumn.
Of the cows included in the study, 3,207 were born in winter, 6,680 in spring, 3,711 in summer and 1,434 in autumn. These seasonal data have been included in Table 1.
Comment: The section on variance components and genetic parameter estimation lacks details about the method used for estimation.
Response: During the work, the BLUP animal model, MTDFREML software, was used to estimate population genetic parameters. The models used are presented in Table 2. In section 2.2, the fixed and random effects built into the model are presented, and the model formula is also shown there.
Comment: Furthermore, were all animals evaluated at the same parity number, and were they primiparous or multiparous?
Response: All of the cows included in the study were first-calf cows, and their ages ranged from 24 to 36 months.
Comment: The Results and Discussion sections have not undergone significant revisions, leaving the implications and limitations of the study unclear.
Response: We have attempted to significantly revise the Results and Discussion sections. This chapter has previously been split into two parts in order to better highlight the implications and limitations. Reorganization and change has been made in several places. Many new sentences have been added to make the implications and limitations more understandable.
Comment: At this stage, I am unable to endorse the manuscript for publication, as it does not meet the journal's standards.
Response: We respectfully request that our reviewer accept our improved manuscript based on our modifications.
Round 3
Reviewer 2 Report
Comments and Suggestions for Authors
I cannot endorse this manuscript for publication, as the flaws in both the methodology and scientific writing remain significant.
The authors claim that BLUP was used to estimate variance components. However, BLUP is not a method for estimating variance components; rather, it is used to estimate random and fixed effects. Typically, variance components are estimated using methods such as the minimum norm quadratic unbiased estimator (MINQUE), analysis of variance (ANOVA), maximum likelihood (ML), restricted maximum likelihood (REML), or Bayesian approaches. This fundamental misunderstanding raises concerns about whether the authors fully grasp the methodology they are employing.
Furthermore, all analytical methods should clearly describe the assumptions, expectations, and limitations of the models used. When constructing a linear mixed model equation, three essential components must be defined:
- The model equation itself;
- The expectations, as well as the variance and covariance structures of the random variables;
- The assumptions, constraints, and limitations.
The last point is particularly crucial, as it pertains to the effects included in the model. Even though these effects are often omitted from discussions, they are vital for ensuring a comprehensive understanding of the model.
Regarding the use of contemporary groups (CG), even some published papers do not handle this correctly. CGs ensure that individuals have an equal opportunity to perform under similar conditions. Additionally, as seen in Henderson’s work, defining CGs based on herd-year-season also accounts for superior sires being placed in better-managed herds, thereby reducing potential biases in the results. I strongly encourage the authors to revisit the foundational literature on BLUP and its development, particularly its application in the dairy industry, to gain a clearer understanding of these principles.
In terms of scientific writing, I do not see sufficient improvements that would allow me to recommend this manuscript for publication. Approving a paper with methodological flaws and weak scientific writing would contradict the standards of this journal and undermine the integrity of the existing scientific literature.
Author Response
Dear Reviewer,
We would like to thank you for your further, constructive comments and suggestions. We will try to consider these when developing our manuscript.
Comment: The authors claim that BLUP was used to estimate variance components. However, BLUP is not a method for estimating variance components; rather, it is used to estimate random and fixed effects. Typically, variance components are estimated using methods such as the minimum norm quadratic unbiased estimator (MINQUE), analysis of variance (ANOVA), maximum likelihood (ML), restricted maximum likelihood (REML), or Bayesian approaches. This fundamental misunderstanding raises concerns about whether the authors fully grasp the methodology they are employing.
Response: Agree that BLUP is not a method for estimating variance components; rather, it is used to estimate random and fixed effects.
Also, agree with the reviewer that it was not clear from the previous manuscript that we did not use the BLUP method to estimate variance components and heritability. We apologize for the imprecise wording and the short explanation of the method.
In fact, we used the REML model to estimate the variance components and BLUP for the genetic value, i.e. breeding value estimation. It would not have been possible to estimate the variance components using the BLUP method because it is not suitable.
We fully agree with our esteemed opponent that variance components are estimated using methods such as the minimum norm quadratic unbiased estimator (MINQUE), analysis of variance (ANOVA), maximum likelihood (ML), restricted maximum likelihood (REML), or Bayesian approaches.
For this reason, we used the REML model to evaluate the phenotypic components. However, phenotypic variance data alone would not have been sufficient to assess heritability and genetic correlations. Therefore, we needed the genetic value, i.e. the breeding value, of each animal for each trait. Breeding value was estimated using BLUP model. When we had the genetic value of the same traits, we could estimate the genetic variation, either using the REML model.
Heritability could be estimated from the total phenotypic and genetic variance.
We were able to calculate a phenotypic correlation between the above values of the different traits and a genetic correlation between their breeding values, moreover the phenotypic and genetic trends.
Comment: Furthermore, all analytical methods should clearly describe the assumptions, expectations, and limitations of the models used. When constructing a linear mixed model equation, three essential components must be defined:
The model equation itself;
The expectations, as well as the variance and covariance structures of the random variables;
The assumptions, constraints, and limitations.
The last point is particularly crucial, as it pertains to the effects included in the model. Even though these effects are often omitted from discussions, they are vital for ensuring a comprehensive understanding of the model.
Response: We tried to improve the methodological part, to include the applied models in the manuscript, and to highlight the assumptions, constraints, and limitations.
Comment: Regarding the use of contemporary groups (CG), even some published papers do not handle this correctly. CGs ensure that individuals have an equal opportunity to perform under similar conditions. Additionally, as seen in Henderson’s work, defining CGs based on herd-year-season also accounts for superior sires being placed in better-managed herds, thereby reducing potential biases in the results. I strongly encourage the authors to revisit the foundational literature on BLUP and its development, particularly its application in the dairy industry, to gain a clearer understanding of these principles.
Response: We completely agree that CGs ensure that individuals have an equal opportunity to perform under similar conditions. However, we would have had very limited opportunities to form CGs, which would not have given reliable results due to the relatively small numbers. For this reason, we chose the BLUP Animal model, which, to our knowledge, allows for unbiased prediction by handling fixed effects.
Comment: In terms of scientific writing, I do not see sufficient improvements that would allow me to recommend this manuscript for publication. Approving a paper with methodological flaws and weak scientific writing would contradict the standards of this journal and undermine the integrity of the existing scientific literature.
Response: Based on the comments we have tried to improve the scientific writing.
We have made many corrections and changes to the introduction, materials and methods and discussion chapters.
We trust that the introduction section is now adequate, as it has not been criticized. The methodological section has been substantially expanded on the basis of above, and the discussion section has also been modified. Taking into account the reviewer's opinion, a few sentences on the methods used have been added to the discussion chapter.
We respectfully request that our opponent accept the revised version of our manuscript.
